# The Cakewalk Method

## Abstract

Combinatorial optimization is a common theme in computer science. While in general such problems are NP-Hard, from a practical point of view, locally optimal solutions can be useful. In some combinatorial problems however, it can be hard to define meaningful solution neighborhoods that connect large portions of the search space, thus hindering methods that search this space directly. We suggest to circumvent such cases by utilizing a policy gradient algorithm that transforms the problem to the continuous domain, and to optimize a new surrogate objective that renders the former as generic stochastic optimizer. This is achieved by producing a surrogate objective whose distribution is fixed and predetermined, thus removing the need to fine-tune various hyper-parameters in a case by case manner. Since we are interested in methods which can successfully recover locally optimal solutions, we use the problem of finding locally maximal cliques as a challenging experimental benchmark, and we report results on a large dataset of graphs that is designed to test clique finding algorithms. Notably, we show in this benchmark that fixing the distribution of the surrogate is key to consistently recovering locally optimal solutions, and that our surrogate objective leads to an algorithm that outperforms other methods we have tested in a number of measures.

## 1 Introduction

Combinatorial optimization is one of the foundational problems of computer science. Though in general such problems are NP-hard Papadimitriou (2003), it is often the case that locally optimal solutions can be useful in practice. In clustering for example, a common objective is to divide a given set of examples into a fixed number of groups in a manner that would minimize the distances between group members. As enumerating all the possible groupings is usually intractable, local search methods such as k-means MacQueen et al. (1967) are frequently used to approach such problems. We find the persistent use of k-means in a wide variety of applications as convincing evidence that from a practical perspective, locally optimal solutions can be useful.

In the combinatorial setting however, solution neighborhoods are not always available, and even when they are, in many interesting cases they only connect small parts of the search space. For example, when the search space involves computer programs, it is not clear how replacing one operation with another (for example, an if clause with an addition operation) impacts the program behavior even if the program validity is preserved. Though one can define a limited but sensible set of neighboring solutions (e.g., replace an addition with a multiplication), neighborhoods that build on those usually connect only a tiny fraction of the search space. Another interesting case involves natural language sentences where replacing one word with another (say, 'very' to 'not'), or changing clauses order can completely change the meaning of a sentence. A third popular scenario involves sequential decision making as is the case in reinforcement learning problems with discrete action spaces, where it is not always clear that two action sequences can be related if the initial actions are different. In such combinatorial problems, methods that transforms one solution to another (either directly or through smoothing) might be confined to a small sub-space, and therefore in such problems, searching the solution space directly is unfavorable.

One type of algorithms which are suitable to such combinatorial problems, and have drawn considerable interest in the last few years are policy gradient methods Sutton & Barto (2017). The general strategy these methods adopt is to construct a parametric sampling distribution over the search space, and to optimize the expected value of some given objective function by applying gradient updates in the parameters' space. In spite of their apparent generality, these gradient updates

require special attention. In particular, the sampled objective values affect both the sign and the magnitude of the gradient step size. On the one hand, such dependence on the objective values is what allows these algorithms to give higher likelihood to examples which achieve better objective values. On the other hand, such direct dependence makes it hard to tune the step sizes by means of predetermined hyper-parameters. As our goal is to extend such constructions to any objective in a generic fashion, we seek to transform the construction so that it will only be sensitive to the order relation the objective induces. In this construction however, the objective is essentially a random variable whose distribution changes from one problem to another, and not only that, it keeps on changing throughout the optimization. As a result, it seems that finding a generic rule for tuning various hyper-parameters in a manner that fits all scenarios seems impractical.

Following this understanding, we purpose to utilize a generic surrogate objective function that has the following two properties. First, the surrogate should preserve the set of locally optimal solutions if solution neighborhoods can be defined. Second, the surrogate should have a fixed and predetermined distribution for every possible objective, and this distribution should remain fixed throughout the optimization. Once in this form, generic rules for setting various hyper-parameters can be found, and that can provide us with a generic stochastic optimizer. Though it might seem that such general purpose surrogate objectives could be hard to find, we show that by utilizing the empirical cumulative distribution function (CDF henceforth) of the original objective, these can be easily constructed. We discuss few possible surrogate objectives, and purpose one such version which makes the basis our method. Since the crux of our method is based on capitalizing on the CDF of the original objective, we refer to our method as CAkEWaLK which stands for CumulAtivEly Weighted LiKelihood.

We start by considering policy gradient methods as stochastic optimization algorithms for combinatorial problems in section 2, and proceed to present the Cakewalk method in section 3. In section 4 we discuss how Cakewalk is related to the cross-entropy (CE henceforth) method, to policy-gradient methods in reinforcement learning, and to multi-arm bandit algorithms. Since we are interested in methods that can recover locally optimal solutions when these can be defined, we use the problem of finding inclusion maximal cliques in undirected graphs as a controlled experiment for testing this property in a non-trivial setting. For that matter, in section 6 we investigate how to apply such methods to the clique problem, and in section 7 we report experimental results on a dataset of graphs on which results are regularly published. Lastly, as an additional experimental task, we show in appendix section B how Cakewalk can be used to produce an algorithm that outperforms the most commonly used algorithms for k-medoids clustering, the combinatorial counterpart of k-means. Notably, we use this task to demonstrate that Cakewalk can also be used to optimize the starting point of greedy algorithms that search the input space directly, thus providing empirical evidence that supports Cakewalk's effectivity in a greater variety of combinatorial problems.

## 2 BACKGROUND

We set out on constructing a stochastic optimization algorithm for combinatorial optimization problems, and start by stating the problem. Let $f$ be an objective function which we need to maximize, and let $\boldsymbol{x} \in [M]^N$ be a string that describes $N$ items such that each $x_j$ is one of a discrete set of $M$ items. In this text we denote discrete sets $\{1, \ldots, K\}$ using $[K]$ . Our goal is to search the space $\mathcal{X} = [M]^N$ for some $\boldsymbol{x}^\star$ that achieves an optimal $f(\boldsymbol{x}^\star) = y^\star$. Since $\mathcal{X}$ is discrete, in general this problem is NP-Hard (maximum clique can be reduced to this description), hence we focus only on finding locally optimal solutions rather than the global optimum $\boldsymbol{x}^\star$. For the purpose of defining locally optimal solutions, we'll rely on a neighborhood function $\mathcal{N}$ that maps each $\boldsymbol{x}$ to its neighboring set, though the methods we consider treat $f$ as a black-box, and don't require such $\mathcal{N}$ for their operation. Our goal is to find some locally optimal solution $\boldsymbol{x}^* \in \mathcal{X}_f^*$ where the set of locally optimal solutions is defined as $\mathcal{X}_f^* = \{\boldsymbol{x} \in \mathcal{X} | \forall \boldsymbol{x}' \in \mathcal{N}(\boldsymbol{x}) . f(\boldsymbol{x}) \geq f(\boldsymbol{x}')\}$. Preferably, we would like to find some $\boldsymbol{x}^*$ whose objective value $y^* = f(\boldsymbol{x}^*)$ is as large as possible, though in general, this cannot be guaranteed.

We describe a stochastic optimization algorithm for problems of this form. Let $\boldsymbol{X}$ be a random variable that is defined over $\mathcal{X}$, and which is distributed according to a parametric distribution $\mathbb{P}_{\boldsymbol{\theta}}$ that the algorithm maintains. In addition, let $\boldsymbol{Y}$ be a random variable that is defined over the values of the objective function $f$, i.e. $\boldsymbol{Y} = f(\boldsymbol{X})$. We emphasize that in this text we refer to random variables using capital English letters in bold such as $\boldsymbol{X}$ or $\boldsymbol{Y}$, and we use $\boldsymbol{x}$ and $y$ to refer to elements in

their appropriate sample spaces (deterministic quantities). During the optimization, the algorithm iteratively samples solutions $\boldsymbol{x}$ according to $\mathbb{P}_{\boldsymbol{\theta}}$, and updates the parameters $\boldsymbol{\theta} \in \mathbb{R}^d$ which govern $\mathbb{P}_{\boldsymbol{\theta}}$ in a manner that reflects how good is the objective value $y = f(\boldsymbol{x})$ with which $\boldsymbol{x}$ is associated. Initially $\mathbb{P}_{\boldsymbol{\theta}}(\boldsymbol{X} = \boldsymbol{x})$ is multi-variate uniform (fully specified in section 5), but as the optimization continues, the algorithm decreases the entropy in the distribution until eventually only few solutions become likely, and sampling some $\boldsymbol{x}$ from $\mathbb{P}_{\boldsymbol{\theta}}$, with high probability, returns some locally optimal solution. Since we discuss an iterative algorithm that at each iteration $t$ updates the parameters $\boldsymbol{\theta}_t$, we refer to the random variables that are associated with $\mathbb{P}_{\boldsymbol{\theta}_t}$ by $\boldsymbol{X}_t$ and $\boldsymbol{Y}_t$. Lastly, as a short hand notation, we refer to $\mathbb{P}_{\boldsymbol{\theta}}(\boldsymbol{X} = \boldsymbol{x})$ simply by $\mathbb{P}_{\boldsymbol{\theta}}(\boldsymbol{x})$.

Since we learn a distribution function, we say that our learning objective $J(\boldsymbol{\theta})$ is to maximize the expectation over $\boldsymbol{x} \sim \mathbb{P}_{\boldsymbol{\theta}}$ of the original objective which we denote as $\mathbb{E}_{\boldsymbol{\theta}}[\boldsymbol{Y}]$. To find the parameters $\boldsymbol{\theta}$ which maximize $\mathbb{E}_{\boldsymbol{\theta}}[\boldsymbol{Y}]$, we derive a gradient ascent algorithm which relies on estimates of $\nabla_{\boldsymbol{\theta}}\mathbb{E}_{\boldsymbol{\theta}}[\boldsymbol{Y}]$. To calculate the gradient, we use the log-derivative trick, $\nabla_{\boldsymbol{\theta}}\mathbb{E}_{\boldsymbol{\theta}}[\boldsymbol{Y}] = \mathbb{E}_{\boldsymbol{\theta}}[\boldsymbol{Y}\nabla_{\boldsymbol{\theta}}\log\mathbb{P}_{\boldsymbol{\theta}}(\boldsymbol{X})]$. Next, we can use Monte Carlo estimation Wasserman (2013) to estimate $\mathbb{E}_{\boldsymbol{\theta}}[\boldsymbol{Y}\nabla_{\boldsymbol{\theta}}\log\mathbb{P}_{\boldsymbol{\theta}}(\boldsymbol{X})]$. Traditionally, at each iteration $t$, a large sample $S_t = \{\boldsymbol{x}_t^k, y_t^k\}_{k=1}^K$ of some fixed size $K$ is sampled using $\mathbb{P}_{\boldsymbol{\theta}_t}$. Denoting this estimate by $\boldsymbol{\Delta}_t$, then the update at iteration $t$ takes the following form,

$$\boldsymbol{\theta}_t = \boldsymbol{\theta}_{t-1} + \eta_t\boldsymbol{\Delta}_t \tag{1}$$

$$\boldsymbol{\Delta}_t = \frac{1}{K}\sum_{k=1}^K \left[ y_t^k \nabla_{\boldsymbol{\theta}} \log \mathbb{P}_{\boldsymbol{\theta}}(\boldsymbol{x}_t^k) \right] \tag{2}$$

where $\eta_t$ is a learning rate parameter that is predetermined. We describe the update step using a vanilla gradient update mostly for illustratory purposes, though in practice any gradient based update such as AdaGrad Duchi et al. (2011) or Adam Kingma & Ba (2014) can be used instead. Turns out that for positive learning rates this stochastic optimization scheme converges to a local maximum of $J$, and when using the optimal parameters $\boldsymbol{\theta}^*$, sampling from $\mathbb{P}_{\boldsymbol{\theta}^*}$ returns locally optimal solutions $\boldsymbol{x}^* \in \mathcal{X}_f^*$ with high probability Williams (1992). Nonetheless, such gradient estimates are known to be highly variable Paisley et al. (2012), which requires drawing large samples at each iteration which is costly. Though there are techniques for reducing the variance of such estimates Ross (2013), these are mostly useful when tied to the specifics of a given objective. We approach this problem differently, and consider instead how can we adapt the optimization objective in a manner that allows us to rely on noisy gradient estimates that only involve a single example (i.e., setting $K = 1$), while ensuring we converge to a distribution that still allows us to sample some $\boldsymbol{x}^* \in \mathcal{X}_f^*$. Since we focus on online updates, for the reminder of the text we drop the superscript $^k$ when referring to $\boldsymbol{x}_t^k$ and $y_t^k$.

## 3 CAKEWALK METHOD

At this point, we've set the stage for discussing how can we transform the previous construction into a generic stochastic optimizer. We start by examining equations 1 and 2 and observing that if we update $\boldsymbol{\theta}_t$ in the direction of $\nabla_{\boldsymbol{\theta}}\log\mathbb{P}_{\boldsymbol{\theta}}(\boldsymbol{x}_t)$, $\eta_t y_t$ can be considered as the step we're taking in that direction. Thus, the sign and magnitude of $\eta_t y_t$ essentially determine whether we increase or decrease the likelihood of $\boldsymbol{x}_t$, and to what extent we do so. The implication this has over the optimization is that the distributions of $\{\eta_t \boldsymbol{Y}_t\}_{t=1}^T$ determine the course of the optimization. If for example $|\eta_t \boldsymbol{Y}_t|$ is unbounded from above, we might take steps that are too large, which might cause us to diverge. Steps that are too small are unfavorable as well, as these will keep the sampling distribution too close to uniform, and due to the combinatorial nature of $\mathcal{X}$, finding good $\boldsymbol{x}$s can take exponentially many examples. This extends to scenarios that involve functions that have a different scale. For example, suppose that we have two functions such that $f_2(\boldsymbol{x}) = cf_1(\boldsymbol{x})$ for every $\boldsymbol{x}$, with $c$ being some fixed positive constant. Clearly, $\mathcal{X}_{f_1}^* = \mathcal{X}_{f_2}^*$, nonetheless, sampling and updating the parameters using equations 1 and 2 would change the speed of the optimization by a factor $c$. Though one can adjust the learning rates to the particularities of some given objective, such an approach would require that we tune the optimization on a case by case basis. Lastly, since in general we don't know ahead of time the distribution of each $\boldsymbol{Y}_t$, it seems that if we follow the construction presented in section 2, we won't be able to determine the series $\{\eta_t\}_{t=1}^T$ in a manner that would fit all scenarios. This reasoning leads us to conclude that if we wish to obtain generic updates, we must come up with some fixed surrogate objective function which preserves $\mathcal{X}_f^*$, and for which we can determine the distributions of

$\{Y_t\}_{t=1}^T$ ahead of time. To achieve this, we suggest a weight function $w$ that when composed over $f$ (i.e. $w \circ f$) produces a surrogate objective that meets these criteria.

Preserving the original set of optimal solution is the easy part, as all we need to do is to require that $w$ will be monotonic increasing, and that would imply that $\mathcal{X}_f^* \subseteq \mathcal{X}_{w \circ f}^*$ (and strict monotonicity would ensure that $\mathcal{X}_f^* = \mathcal{X}_w^*$ though we don't insist on that). The harder part is to construct $w$ in a manner that would fix the distribution of $w(\boldsymbol{Y}_t)$ for all $t$. Nonetheless, basic probability tells us that if $F_t$ is the CDF of $Y_t$, then $F_t(\boldsymbol{Y}_t)$ is uniformly distributed on $[0, 1]$ Wasserman (2013). Since every CDF is monotonic increasing, if we construct $w$ using $F_t$, we can preserve the original set of optimal solutions. More importantly, if we can estimate $F_t$, we could use it to produce our surrogate objective as it **would fix the surrogate's distribution once and for all**, thus making significant progress towards our goal. Next, since insisting that $w(Y_t) \sim U(0, 1)$ might not be ideal, we take this idea one step further, and utilize another monotonic increasing function $g$ for which $g(F_t(\boldsymbol{Y}_t))$ can be distributed differently. For purposes that we specify next, we also require that $g$ will be bounded.

Since we don't have access to $F_t$ in general, as was the case with the gradient, we need to estimate it from data. Fortunately enough, since the image of $f$ is one dimensional (an optimization objective), order statistics can supply us with highly reliable non-parametric estimates for each $F_t$. The only question that comes up is how can we perform the aforementioned estimation without drawing a large sample at each iteration. Due to equation 2, if we use a sampling distribution for which $\|\nabla_{\boldsymbol{\theta}} \log \mathbb{P}_{\boldsymbol{\theta}}(\boldsymbol{x}_t)\|$ is bounded, then since $w(y_t)$ is bounded as well, $\|\boldsymbol{\Delta}_t\|$ will be bounded for every $\boldsymbol{x}_t$ and $y_t$. The main implication of this property is that we can control how different the parameters will be between any two iterations, i.e., that for any two iterations $t$ and $t-k$ where $k \in [t-1]$ we can make $\|\boldsymbol{\theta}_t - \boldsymbol{\theta}_{t-k}\|$ as small as we want simply by changing $\eta_t$. Thus, instead of drawing a large sample in each iteration, we can say the last objective values $y_{t-1}, \ldots, y_{t-k}$ are approximately i.i.d from $\mathbb{P}_{\boldsymbol{\theta}_{t-1}}$. Therefore, if we use small enough learning rates, we can use $\hat{F}_{t-1}(y) = \frac{1}{k} \sum_{i=1}^k \mathbb{I}[y_{t-i} < y]$ as an estimator for $F_{t-1}$, where $\mathbb{I}[\cdot]$ is the indicator function. In our experiments, using some fixed learning rate $\eta \in (0, 1)$ along with $k = \frac{1}{\eta}$ seem to work quite well. Overall, the parameters' updates we suggest have the following form,

$$\boldsymbol{\Delta}_t = g\left(\hat{F}_{t-1}(y_t)\right) \nabla_{\boldsymbol{\theta}} \log \mathbb{P}_{\boldsymbol{\theta}}(\boldsymbol{x}_t) \tag{3}$$

### 3.1 SURROGATE OBJECTIVES

In this section, we focus on a single iteration $t$, and thus, drop the subscript $t$ in all cases. The purpose of the first option we present is to illustrate the connection between our algorithm, and the CE method, and for that reason we denote this weight function by $\hat{w}_{CE}(y) = g_{CE}\left(\hat{F}(y)\right)$, and its associated transformation by $g_{CE}$. Given some small $\rho \in [0, 1]$ which is decided by the user a-priori (typically, 0.1 or 0.01), $g_{CE}$ is a thresholding function $g_{CE}(z) = \mathbb{I}[z \geq 1 - \rho]$. Clearly, for any fixed $\rho$, $\hat{w}_{CE}$ is monotonic increasing and bounded, and $\hat{w}_{CE}(\boldsymbol{Y})$ is a Bernoulli random variable with probability $\rho$. Notably, using $g_{CE}$ in equation 3 leads to an update which can be considered as an online version of the CE method. There are two main disadvantages to $\hat{w}_{CE}$. First of all, it relies on another parameter $\rho$ that requires manual tuning. More importantly, $\hat{w}_{CE}$ uses only the highest $\rho$ percentile of the examples to update $\mathbb{P}_{\boldsymbol{\theta}}$ while in fact the worst $\boldsymbol{x}$s supply valuable information - they have low objective values, and thus, their likelihood should be decreased rather than simply ignored. Thus, we suggest two weight functions which fix these issues. Probably the simplest option is to use the empirical CDF $\hat{F}$ directly, which would make $\hat{F}(\boldsymbol{Y})$ uniform discrete on $[0, 1]$. While this surrogate doesn't involve any extra parameters, nor does it ignore the information supplied by every $\boldsymbol{x}$, it still has one major drawback, it leads to an increase in the likelihood of every example it sees. This create a bias towards $\boldsymbol{x}$s that have already been sampled, compared with $\boldsymbol{x}$s that weren't, even though their associated objective value might be better. Since $\mathcal{X}$ grows exponentially fast with $N$, as $N$ grows, examples that are drawn early in the process can influence the course of the optimization dramatically. Following this reasoning, we adjust $\hat{F}$ so that it would only increase the likelihood of only half of the examples, and decrease the likelihood of the other half. To do so, we make $\hat{w}(y) = 2\hat{F}(y) - 1$. By construction, it follows that $\hat{w}(\boldsymbol{Y})$ is uniform discrete on $[-1, 1]$. In this fashion, when applied with some fixed learning rate, $\hat{w}$ determines whether the likelihood of

---

**Algorithm 1** Cakewalk

---

**input** $f$, $\mathbb{P}_{\boldsymbol{\theta}}$, $k$, $Add$ {objective function $f$, sampling distribution $\mathbb{P}_{\boldsymbol{\theta}}$, integer $k$, gradient addition rule $Add$}
initialize $\boldsymbol{\theta}_0$
**repeat** {for every $t$ in $1, 2, \ldots$ }
   $\boldsymbol{x}_t \sim \mathbb{P}_{\boldsymbol{\theta}_{t-1}}$ {sampling an example}
   $y_t = f(\boldsymbol{x}_t)$ {objective function evaluation}
   **if** $t > k$ **then**
      $w_t = 2\left(\frac{1}{k}\sum_{i=1}^{k}\mathbb{I}\left[y_{t-i} < y_t\right]\right) - 1$
      $\boldsymbol{\Delta}_t = w_t \nabla_{\boldsymbol{\theta}} \log \mathbb{P}_{\boldsymbol{\theta}}(\boldsymbol{x}_t)$
      $\boldsymbol{\theta}_t = Add(\boldsymbol{\theta}_{t-1}, \boldsymbol{\Delta}_t)$
   **end if**
**until** convergence
**return** $\boldsymbol{x}^*$ which had the highest $y^*$

---

some example will be increased or decreased, and to what extent. Notably, this is achieved along with full specification of the distribution of $\hat{w}(\boldsymbol{Y})$. This is a major advantage compared with, for example, transforming $\boldsymbol{Y}$ with its estimated z-score, as in this case we can't determine how $w(\boldsymbol{Y})$ is distributed, nor can we guarantee that $|w(\boldsymbol{Y})|$ is bounded (leading to a risk of divergence, and disrupting of the online estimation of $F$). We summarize Cakewalk with $\hat{w}$, and any gradient addition rule $Add$ (this includes hyper-parameters such as learning rate) in algorithm 1.

## 4 RELATED WORK

Our method is closely related to the CE method. CE was introduced by Rubinstein initially for estimating low probability events Rubinstein (1997), and later adapted to combinatorial optimization problems Rubinstein (2001). Turns out that when CE is applied with discrete sampling distributions the likelihood-ratio term cancels out, and the construction is equivalent to maximizing the likelihood of the examples whose objective values belong to the highest $\rho$ percentile De Boer et al. (2005). Thus, in this case CE's update step is equivalent to maximizing the surrogate objective $\hat{w}_{CE}$ described in section 3.1. As discussed in section 3.1, $\hat{w}_{CE}$ has two major shortcomings, and these lead us to suggest a different surrogate objective which makes the basis for Cakewalk. In addition to these differences, Cakewalk is an online algorithm whereas CE requires drawing a large sample in each iteration so as to estimate the CDF. Our construction enables us to rely on bounded gradient updates that facilitate online estimation of the CDF, and therefore Cakewalk's iterations are considerably less computationally expensive than those of CE. The next family of algorithms to which our method is related to are policy gradient methods. The research on these was initiated by Williams with REINFORCE Williams (1988), an algorithm which we consider as the prototype to Cakewalk, and which provides Cakewalk with convergence guarantees. Most of the work on policy gradient methods derives from REINFORCE, essentially discussing how to rescale the objective in various scenarios. For example, actor critic methods Sutton & Barto (2017) use estimates $\hat{\mu}$ of $\mathbb{E}(\boldsymbol{Y})$ that are produced with some model of the objective, and can be used to make $y - \hat{\mu}$ zero mean. As these methods rely on a particular model of the objective, they are inherently problem specific. Of these methods, probably the natural actor-critic algorithm Peters & Schaal (2008) better fits Cakewalk's general purpose nature. This algorithm rescales the estimated gradient by multiplying it by the inverse of the Fisher information matrix. As this requires large sample to accurately estimate both the gradient, and of the Fisher information matrix, the natural actor-critic is considerably more computationally expensive than online algorithms such as Cakewalk or REINFORCE. The third family of algorithms to which our method is related are multi-arm bandit algorithms. In the bandit setting, a learner is faced with a sequential decision problem, where in each round an arm is chosen, and each arm is associated with some non-deterministic loss. Initially suggested by Thompson Thompson (1933), this setting has been explored extensively with the notable successes of the UCB algorithm Auer (2002); Auer et al. (2002a) for cases where the losses are stochastic, and the Exp3 Auer et al. (1995; 2002b) for when they can even be determined by an adversary. Over the years these have become a basis for a wide variety of algorithms Bubeck et al. (2012) for various settings which that even extend to cases that involve high dimensional structured arms Awerbuch & Kleinberg (2004); McMahan & Blum

Table 1: Local optimality, Hamming neighbourhood, higher is better.[1]

|  | Exp3 | REINF | REINF$_B$ | REINF$_Z$ | OCE$_{0.01}$ | OCE$_{0.1}$ | CW $\hat{F}$ | CW $\hat{w}$ |
|---|---|---|---|---|---|---|---|---|
| SGA | 0.000* | 0.001* | 0.001* | 0.097 | 0.001* | 0.000* | 0.002* | 0.042 |
| AdaGrad | 0.000* | 0.000* | 0.352* | 0.427* | 0.077* | 0.691* | 0.164* | **0.835** |
| Adam | 0.000* | 0.000* | 0.525* | 0.616* | 0.106* | 0.353* | 0.184* | 0.753 |

Table 2: Local optimality, inclusion maximality, higher is better.

|  | Exp3 | REINF | REINF$_B$ | REINF$_Z$ | OCE$_{0.01}$ | OCE$_{0.1}$ | CW $\hat{F}$ | CW $\hat{w}$ |
|---|---|---|---|---|---|---|---|---|
| SGA | 0.000 | 0.000 | 0.000 | 0.138 | 0.000 | 0.000 | 0.000 | 0.037 |
| AdaGrad | 0.000* | 0.000* | 0.637* | 0.688* | 0.063* | 0.875 | 0.175* | **0.912** |
| Adam | 0.000* | 0.000* | 0.662* | 0.787 | 0.100* | 0.412* | 0.212* | 0.887 |

(2004); Cesa-Bianchi & Lugosi (2012). The key difference between the bandit and the optimization setting is that the losses associated with each of the arms are non-deterministic, and thus in the bandit setting the main challenge is to balance estimating the statistics associated with each of the arms, with exploiting the information gathered thus far. In the optimization setting however, the goal is simply to find the best deterministic solution using the least number of steps. Thus, in spite of the apparent similarity, it is this fundamental difference that separates the optimization from bandit settings.

## 5 SAMPLING DISTRIBUTION

Before we specify a particular distribution, we wish to emphasize that the Cakewalk update rule isn't tied to any particular sampling distribution. The distribution we specify next is only used as an example, and as a basis for the experiments we report later. Following Rubinstein's construction, we use a simple distribution that factorizes into a sequence of independent distributions, each defined over a different dimension. In this manner, the number of parameters required to represent $\mathbb{P}_{\boldsymbol{\theta}}(\boldsymbol{x})$ grows only linearly with $MN$, instead of the exponential number of parameters that is required to represent the full joint distribution. Formally, each $x_j$ is drawn independently according to a softmax distribution $\mathbb{P}_{\boldsymbol{\theta}}(x_j = i) = \frac{e^{\theta_{i,j}}}{\sum_{k \in [M]} e^{\theta_{k,j}}}$ where $i \in [M]$, and therefore $\mathbb{P}_{\boldsymbol{\theta}}(\boldsymbol{x}) = \prod_{j \in [N]} \frac{e^{\theta_{x_j,j}}}{\sum_{k \in [M]} e^{\theta_{k,j}}}$.

Next, we describe $\nabla_{\boldsymbol{\theta}} \log \mathbb{P}_{\boldsymbol{\theta}}(\boldsymbol{x})$ in terms of partial derivatives, $\frac{\partial \log \mathbb{P}_{\boldsymbol{\theta}}(\boldsymbol{x})}{\partial \theta_{i,j}} = \mathbb{I}[x_j = i] - \mathbb{P}_{\boldsymbol{\theta}}(x_j = i)$. Note that since $\|\nabla_{\boldsymbol{\theta}} \log \mathbb{P}_{\boldsymbol{\theta}}(\boldsymbol{x})\|$ is bounded, we can estimate $\hat{F}$ in an online manner.

## 6 STUDYING LOCAL OPTIMALITY USING CLIQUE FINDING

In this section, we set the grounds for investigating whether algorithms that only rely on function evaluations can recover locally optimal solutions. We emphasize that our goal is to investigate this question, and not compete with algorithms that rely on some neighborhood function for their operation, and which search the input space directly. We study this question on a NP-hard problem instead of problem for which we can find the global optimum in polynomial time, as it important to verify that such methods can recover non-trivial optima in challenging scenarios. We focus on the problem of finding locally maximal cliques, as the notion of inclusion maximal cliques naturally entails what neighborhood function should be used to judge this property. Formally, a graph $G$ is a pair $(V, E)$ where $V = [N]$ is a set of vertices, and $E \subseteq V \times V$ is a set of edges. $G$ is undirected if for every $(i, j) \in E$ it follows that $(j, i) \in E$. A clique in an undirected graph is a subset of vertices $U \subseteq V$ such that each pair of which is connected by an edge. An inclusion maximal clique $U$ is such that there is no other $v \in V \setminus U$ for which $U \cup \{v\}$ is also a clique.

Next, we design an objective that could inform algorithms that only rely on function evaluations how densely connected is some subgraph, and which favors larger subgraphs. We refer to this function as

---

[1] Out performance by CW $\hat{w}$ in a statistical significant is denoted by *. This applies to all tables.

Table 3: Best-sample to total-samples ratio, lower is better.

|         | Exp3 | REINF | REINF$_B$ | REINF$_Z$ | OCE$_{0.01}$ | OCE$_{0.1}$ | CW $\hat{F}$ | CW $\hat{w}$ |
|---------|------|-------|-----------|-----------|--------------|-------------|--------------|--------------|
| SGA     | -    | -     | 0.654     | 0.907     | 0.874        | 0.945       | 0.939        | 0.927        |
| AdaGrad | -    | -     | 0.821*    | 0.821*    | 0.966*       | 0.820*      | 0.926*       | 0.657        |
| Adam    | -    | -     | 0.743*    | 0.731*    | 0.835*       | 0.697*      | 0.741*       | **0.619**    |

Table 4: Largest-returned-clique to largest-known-clique ratio, higher is better.

|         | Exp3   | REINF  | REINF$_B$ | REINF$_Z$ | OCE$_{0.01}$ | OCE$_{0.1}$ | CW $\hat{F}$ | CW $\hat{w}$ |
|---------|--------|--------|-----------|-----------|--------------|-------------|--------------|--------------|
| SGA     | 0.000  | 0.000  | 0.000     | 0.135     | 0.000        | 0.000       | 0.000        | 0.038        |
| AdaGrad | 0.000* | 0.000* | 0.538*    | 0.567*    | 0.062*       | 0.738       | 0.161*       | **0.756**    |
| Adam    | 0.000* | 0.000* | 0.577     | 0.657     | 0.091*       | 0.364*      | 0.190*       | 0.737        |

the soft-clique-size function, and denote it by $f_{SCS}$. For our purposes, we say the space $\mathcal{X} = \{0, 1\}^N$ correspond to strings which determine membership in some subgraph $U$. Let $\boldsymbol{x} \in \mathcal{X}$, then for each vertex $j \in V$, we say that $j \in U$ if and only if $x_j = 1$, and accordingly we denote such subgraphs by $U_{\boldsymbol{x}}$. If some $U_{\boldsymbol{x}}$ is a clique, for every $i, j \in U_{\boldsymbol{x}}, i \neq j$ it follows that $(i, j) \in E$, and therefore $\sum_{i,j \in U_{\boldsymbol{x}}, i \neq j} \mathbb{I}[(i, j) \in E] = |U_{\boldsymbol{x}}| (|U_{\boldsymbol{x}}| - 1)$. As a consequence, dividing by the RHS produces a subgraph density term. Next, we add a parameter $\kappa \in [0, 1]$ that rewards larger subgraphs, and which could indicate to an algorithm it should prefer larger subgraph over smaller ones. To achieve this, we change aforementioned denominator to $|U_{\boldsymbol{x}}| (|U_{\boldsymbol{x}}| - 1 + \kappa)$. Lastly, to avoid division by zero for cases $|U_{\boldsymbol{x}}| < 2$, we can wrap the denominator with $\max(\cdot, 1)$. Altogether,

$$f_{SCS}(\boldsymbol{x}, G, \kappa) = \frac{\sum\limits_{i,j \in U_{\boldsymbol{x}}, i \neq j} \mathbb{I}[(i, j) \in E]}{\max(|U_{\boldsymbol{x}}| (|U_{\boldsymbol{x}}| - 1 + \kappa), 1)}$$

To see why higher $\kappa$ can reward larger cliques we focus on the case that $|U_{\boldsymbol{x}}| \geq 2$, and observe that for $U_{\boldsymbol{x}}$ which is clique, when $\kappa = 0$, $f_{SCS}(\boldsymbol{x}, G, 0) = 1$. However, when $\kappa = 1$, $f_{SCS}(\boldsymbol{x}, G, 1) = \frac{|U_{\boldsymbol{x}}| - 1}{|U_{\boldsymbol{x}}|}$, and thus, the larger $U_{\boldsymbol{x}}$ is, the closer this ratio is to 1. In this manner, increasing $\kappa$ gives larger subgraphs a 'boost' compared with smaller one, though it could be that some subgraph which isn't a clique will receive a higher value than some smaller subgraph which is a clique (only for $\kappa = 0$ we get that $f_{SCS}(\boldsymbol{x}, G, 0) = 1$ necessarily means that $U_{\boldsymbol{x}}$ is clique). Empirically, we see that the algorithms we've tested aren't very sensitive to the value of $\kappa$.

## 7 EXPERIMENTAL RESULTS

As a benchmark for clique finding, we used 80 undirected graphs that were published as part of the second DIMACS challenge Johnson & Trick (1996). Each graph was generated by a random generator that specializes in a particular graph type that conceals cliques in a different manner. The graphs contain up to 4000 nodes, and are varied both in their number of nodes and in their edge density. We tested each method on all 80 graphs, letting it maximize the soft-clique-size function using various values of $\kappa$. To determine if a clique is inclusion maximal, since a-priori we don't know which $\kappa$ will lead to such clique, we've executed each method using each of the values $0.0, 0.1, \ldots, 1.0$ as $\kappa$. In each execution, we've executed a method for $100 |V|$ samples (hence runtime is fixed per graph), and at the execution's end, we recorded both the best solution along with its objective value, as well as the sample number in which that solution was found.

In terms of the methods tested, following the discussion on related work, we experimented with the CE method, three versions of REINFORCE, and of the bandit algorithms we've used Exp3. As we focus on online algorithms, for CE, we used the online version that we derived in this work, using two threshold values suggested by Rubinstein, $\rho = 0.1$ and $\rho = 0.01$, and refer to these as OCE$_{0.1}$ and OCE$_{0.01}$, with O standing for online. Next, we've experimented with three versions of REINFORCE. First is the vanilla version, second is a version where the mean $\hat{\mu}$ is subtracted from $y$ as a baseline,

and a third uses the objective's estimated z-score $\frac{y-\hat{\mu}}{\hat{\sigma}}$. We refer to these by REINF, $\text{REINF}_B$, and $\text{REINF}_Z$. For Cakewalk, we used both the unscaled empirical CDF $\hat{F}$, and its scaled counterpart $\hat{w}$, denoting these as CW $\hat{F}$ and CW $\hat{w}$. Note however that the former is only used for comparisons, and that we identify Cakewalk with the latter. For estimating $\hat{\mu}, \hat{\sigma}$ and $\hat{F}$, we've used the last 100 objective values, and thus, both $\text{REINF}_B$, and $\text{REINF}_Z$ make for important comparison as these only transform the objective values, but do not fix its distribution as CE and Cakewalk do. For gradient update methods, we've used vanilla stochastic gradient ascent (SGA henceforth), AdaGrad, and the Adam updates. The latter two methods are considered scale invariant, and thus could help Exp3, REINF, and $\text{REINF}_B$ handle changes in the objective's scale. Altogether, we've tested 8 optimization methods, 3 update steps, on 80 graphs, and 11 values of $\kappa$, leading to a total of 21120 separate executions. We specify the complete experimental details in the appendix section A.

We analyzed 4 performance measures for each of the 8 optimizers, and the 3 gradient update types, and accordingly report results in four $3 \times 8$ tables. In the following, we refer to each combination of an optimizer and gradient update as a method. First, we examined whether a locally optimal solution was found. To test for local optimality for the soft-clique-size, given a result $x$ in some graph, we compared it to every other $x'$ for that graph whose Hamming distance from $x$ is 1, and checked that no $x'$s achieved higher soft-clique-size. We report average local optimality in such Hamming neighborhoods in table 1. Then, to test inclusion maximality of the returned solutions, since the soft-clique-size doesn't guarantee convergence to cliques, for every graph, we tested whether a method returned at least one inclusion maximal clique when applied with some $\kappa$. We report average inclusion maximality in table 2. Next, since some methods find their best solution earlier than others, to analyze the sampling efficiency of each method, we calculated the ratio of the best sample number and the total number of samples used in that execution. Since this comparison only makes sense when controlling for the quality of the solution, we excluded REINF and Exp3 from it as they didn't return locally optimal solutions. We report average best-sample to total-samples ratio in table 3. To ensure returned solutions aren't trivial (say cliques of size 2), for each graph, we compared the largest inclusion maximal clique found by that method, and compared it to the best known solution for that graph, using results from Nguyen (2017). We report average largest-found-clique to largest-known-clique ratios in table 4. Lastly, we performed multiple hypothesis tests to compare every optimizer to CW $\hat{w}$ in all the experimental conditions using one sided sign test Gibbons & Chakraborti (2011), and to control the false discovery rate Wasserman (2013), we determined the significance threshold at a level of $10^{-2}$ using the Benjamini-Hochberg method Wasserman (2013). The best optimizer in each table is marked using bold fonts.

## 8 DISCUSSION AND CONCLUSIONS

The results in tables 1 and 2 clearly support our main proposition that in the considered setting, a surrogate objective whose distribution is fixed and predetermined significantly improves the rate in which locally optimal solutions are recovered. Both CW $\hat{w}$ and $\text{OCE}_{0.1}$ rely on such surrogates, and both outperform Exp3 and all versions of REINFORCE which do not employ such surrogates. Interestingly, it appears that having a surrogate whose distribution is fixed is more effective than to normalize the objective values as the previous comparison also includes $\text{REINF}_Z$. Nonetheless, not all distributions are as effective ($\text{OCE}_{0.01}$ and CW $\hat{F}$ didn't perform as well), and of the ones that we have tested, uniform on $[-1, 1]$ seems to be favorable. CW $\hat{w}$ clearly outperforms $\text{OCE}_{0.1}$ in table 1, and the latter only comes close in the more permissive comparison which selects the best result out of 11 different executions (different values of $\kappa$) as reported in table 2. In terms of sample efficiency, the results in table 3 show that even though $\text{OCE}_{0.1}$ can recover locally optimal solutions, it is not as efficient as CW $\hat{w}$ which finds the best solution considerably faster. When considering the various gradient updates, it appears that CW $\hat{w}$ with AdaGrad produces the best combination as it outperforms all others methods in almost all measures (CW $\hat{w}$ with Adam converges slightly faster, though at the cost of worse optimality rates). Lastly, the comparisons to the best known results in table 4 show that the recovered solutions are far from trivial, and that Cakewalk might even approach the performance of problem specific algorithms which have access to a complete specification of the problem. Overall, we find these results are a strong indication that Cakewalk is a highly effective optimization method, and we believe that future research will prove its effectiveness in other domains such as continuous non-convex optimization, and in reinforcement learning problems.

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

## A  EXPERIMENTAL SETUP DETAILS

As a benchmark, we used 80 undirected graphs that were published as part of the second DIMACS challenge (Johnson & Trick, 1996) which specifically focused on combinatorial optimization, and included instances of the clique problem. Over the years, this dataset has become a standard benchmark for clique finding algorithms, and results on it are regularly published.

In terms of the methods we use for comparison, of the bandits family of algorithms, we considered Exp3 as more suitable than UCB due to the multi-dimensionality of the problem. For example, adding an isolated vertex $v$ to a set of vertices $U$ who is a clique will damage the objective. Due to such cases, we used Exp3 instead of UCB. We applied Exp3 to each of the $N$ elements independently. Note that the assumption of bounded losses/gains that the Exp3 algorithm is dependent upon is met by the soft-clique-size function.

For the gradient updates, we used SGA, AdaGrad, and Adam. We note that AdaGrad is particularly suited to our setting as applying it on indicator data is one of its classical use cases ($\mathbb{I}\left[x_j = i\right]$ can be considered as our data). Adam on the other hand has proven as effective for training neural networks in a wide variety of problems, and nowadays is probably the mostly commonly used gradient update. We decided to experiment with Exp3 in conjunction with AdaGrad and Adam even though this revokes the theoretical guarantees of Exp3 for completeness purposes. We applied AdaGrad with $\delta = 10^{-6}$, and Adam with $\beta_1 = 0.9, \beta_2 = 0.999, \epsilon = 10^{-6}$. We used a fixed learning rate of $0.01$ in all the executions. All the algorithms were implemented in Julia (Bezanson et al., 2017) by the authors.

## B  APPLICATION TO K-MEDOIDS

As mentioned in the introduction, clustering is a classical problem in which practitioners regularly rely on optimization methods that return locally optimal solutions. For that matter, in this section we study how to apply Cakewalk to the k-medoids (Hastie et al., 2009) problem, the combinatorial counterpart of k-means. As in the k-means, we're given a set of $m$ data points from some input space,

and our goal is to divide these into $k$ clusters in a manner that would minimize their distances to one of $k$ representatives. In k-means, each representative can be any point in the input space, and in k-medoids, the representatives are a subset of original points that we're given. Since in k-medoids the representatives are known in advance, it is enough to consider as input a distance matrix $D \in \mathbb{R}_+^{m \times m}$ where $D_{i,j}$ is the distance between point $i$ and $j$, and $\mathbb{R}_+$ is the set of non-negative reals. Thus, one can think of the problem as selecting $k$ representatives from the $m$ data points, and in the general case where we allow points to represent more than one cluster, the solution space becomes $[m]^k$. Given a set of representatives $x \in [m]^k$, each point $i$ is assigned to the representative $x_j$ which minimizes the distance $D_{i,x_j}$ to it. In this formulation, the k-medoids optimization problem can be stated as follows,

$$\underset{x \in [m]^k}{\text{minimize}} \quad \sum_{i=1}^{m} \left[ \min_{j \in [k]} \left\{ D_{i,x_j} \right\} \right]$$

Since the problem is combinatorial, going over all the possible solutions quickly becomes intractable, and greedy algorithms are usually used to approach the problem. Of these, probably the two most commonly used algorithms are the Voronoi iteration (Hastie et al., 2009), and the more computationally expensive, Partitioning Around Medoids (PAM henceforth) (Kaufman & Rousseeuw, 2009). In both methods, first some initial set of representatives is determined, and the appropriate cluster assignments are determined. In the former method, in each iteration, we seek to replace each representative with some other cluster member, and in the latter we seek to replace each representative with any non-representative point. After the new representatives are determined, cluster assignments are determined, and the process is then repeated as long as the objective is improved. In spite of the obvious computational benefits of the Voronoi iteration, PAM is probably more commonly used as it is known to achieve lower objective values.

Since both methods are greedy, the objective to which they converge is determined by how they are initialized (the algorithms are deterministic). Thus, we can try to find the a good initialization for such greedy algorithms with some optimization algorithm. Since Cakewalk only relies on function evaluations, it doesn't matter if we let it optimize some function $g : \mathcal{X} \to \mathbb{R}$, or a composition $g \circ f$ where $f$ is some deterministic transformation $\mathcal{X} \to \mathcal{X}$ of inputs. As long as some input $\boldsymbol{x}$ is associated with some fixed objective value $y = g\left(f\left(\boldsymbol{x}\right)\right)$, any of the methods discussed earlier will be able to optimize such an objective. The only detail that requires attention is that now instead of returning the best $\boldsymbol{x}^*$ associated with the optimal $y^* = g\left(f\left(\boldsymbol{x}^*\right)\right)$, we'll need to return $f\left(\boldsymbol{x}^*\right)$. In terms of implementation, we can do this by either keeping $f\left(\boldsymbol{x}^*\right)$ instead of $\boldsymbol{x}^*$, or by applying $f$ to the $\boldsymbol{x}^*$ which is returned by Cakewalk. In this manner, optimization algorithms that only rely on function evaluations, and greedy algorithms can come together to produce powerful algorithms that outperform the components that make them up.

### B.1    Experimental Results

To test the effectivity of each of the aforementioned optimizers on the k-medoids problem we've setup the following experiment. Using datasets that are publicly available on White (2017), we collected 38 datasets that had between 500 and 1000 data points, and which had numerical attributes. In order to transform these to a valid input for a k-medoid algorithm, for each dataset, we extracted all the numerical attributes, and used them as a numerical vector that represents some data point. Then, we calculated Mahalanobis distance (Bishop, 2006) between each pair of points, which resulted in a distance matrix for that dataset. For the Mahalanobis distances, we used diagonal covariance matrices. As this point, we were able to run each of the aforementioned algorithms on these datasets. Specifically, we used both the Voronoi iteration and PAM algorithms, as well as vanilla Cakewalk. In order to see if we can combine Cakewalk with a greedy method to produce a combined algorithm that is more powerful, we also used Cakewalk with the Voronoi iteration using the setup mentioned earlier. We decided to use the Voronoi iteration instead of PAM, as in our experiments the former was considerably more efficient. We used Cakewalk with AdaGrad (Duchi et al., 2011) using the same hyper-parameters as specified in section A, except for the learning rate $\eta$, which was set to 0.02 instead of 0.01 as we've seen that it led to faster convergence than the latter. We used these hyper-parameters both when applying Cakewalk alone, and when applying the Cakewalk-Voronoi combination. As a convergence criterion for Cakewalk, we use two exponentially running averages of the objective values, and determined convergence has occurred when their absolute difference ratio was smaller than 0.01. Each running average was produced using a time constant that was calculated

using the following formula $1 - \exp\left(\frac{\ln(a)}{b}\right)$ with $a$ always being $0.01$, and $b$ being a parameter that is adjusted to the size of the problem. Thus, for each dataset with $m$ data points and $k$ clusters, we've set $b = \max(mk, 1000)$ for the short running average, and $2b$ for the long running average. We used the same converge criterion for Cakewalk+Voronoi. Altogether, this provided us with 4 clustering algorithms. All methods we're implemented in Julia by the authors.

As a benchmark, we executed each algorithm on all datasets with $k = 10$, and recorded the smallest objective value that was returned, as well as the number of objective function evaluations that were performed. Since the Voronoi iteration does not fully reevaluate the objective completely after every step (only within each cluster), we didn't record the latter measurement for it. Nonetheless, Cakewalk, PAM, and Cakewalk+Voronoi fully evaluate the objective in each step, and therefore can be compared in terms of their total number of function evaluations.

In the analysis our goal was to produce a ranking of the tested algorithms in terms of best objective values that were found. Since it could be some method achieved a better objective by performing more function evaluations, we also ranked the different algorithm in terms of their total number of function evaluations. Thus, in the following, we refer either to the best objective value, or to the total number of function evaluations as a measurement. To determine the best to worst order of each of the 4 algorithms, we first calculated the ratio between the measurement achieved on some dataset, and the minimal value achieved by any of the algorithms on this dataset. This is important so as to make the ranking invariant to the specifics of each dataset by fixing their scale. Then, we calculated the median of the scaled measurements for each of the four algorithms. This produced 4 values for the objective values, and 3 for the function evaluations. Then, we sorted these to determine the best to worst order. Next, to see if the differences between any two algorithms in some measurement were statistically significant, we validated their order using a one sided sign test (Gibbons & Chakraborti, 2011), applying it directly to the original measurements (unscaled). This procedure produced 3 p-values for the objective values, and two for the function evaluations. Next, to control the false discovery rate (Wasserman, 2013), we determined the significance threshold at a level of $10^{-2}$ using the Benjamini-Hochberg method (Wasserman, 2013). Following this analysis, the best to worst algorithm in terms of objective value (smallest to largest) was as follows,

$$\text{Cakewalk+Voronoi} \overset{*}{<} \text{PAM} \overset{*}{<} \text{Cakewalk} \overset{*}{<} \text{Voronoi}$$

where A $<$ B means that A achieved smaller value than B, and $*$ means that the difference between the two is statistically significant. Next, the best to worst algorithms algorithm in terms of the number of objective function evaluation (smallest to largest, excluding the Voronoi iteration) is as follows,

$$\text{Cakewalk+Voronoi} \overset{*}{<} \text{PAM} < \text{Cakewalk}$$

## B.2 Conclusions

Following the analysis presented in section B.1, we conclude that combining Cakewalk with a greedy algorithm produces a clustering method that outperforms the two most commonly used algorithms for the k-medoids problem. Notably, here we combined Cakewalk with the Voronoi iteration, the weaker of the two in terms of performance, and that already produced a method that outperforms PAM. This suggests that probably combining Cakewalk with PAM can produce an even better clustering method, though we leave this to future research. Furthermore, it seems that applying Cakewalk without any greedy method already outperforms the Voronoi iteration, showing that vanilla Cakewalk can outperform some greedy algorithms as these might be limited by the neighborhood function they rely on, a limitation that doesn't apply to a sampling algorithm such as Cakewalk. In terms of function evaluations, it appears that PAM and Cakewalk perform about the same number of function evaluations (the difference is not statistically significant), and both perform more evaluations than the combination of Cakewalk+Voronoi. Taken together, these results not only show that combining Cakewalk with a greedy method can produce an optimizer that outperforms the components that make it up, it also leads to a combined algorithm that converges faster.

