# OpenReview forum: "The Cakewalk Method"
_ICLR.cc/2019/Conference_

### Official Review · AnonReviewer2 · 2018-11-01
**Cakewalk is too similar to the cross-entropy method to warrant acceptance**

**Rating:** 4
**Confidence:** 4

**Review:**

## Summary ##

The authors apply policy gradients to combinatorial optimization problems. They suggest a surrogate reward function that mitigates the variance in the reward, and hence the update size. They demonstrate performance on a clique-finding problem.


## Assessment ##

I don't think Cakewalk is different enough from the cross-entropy method to warrant acceptance in ICLR.

 I also have concerns about the independence assumption in their sampling distribution (Section 3.2), and the fact that their experiments use the same set of (untuned) hyperparameters for each method.

They both approximate the reward CDF from K samples and use this to construct a surrogate reward. The difference is that Cakewalk uses the CDF directly, while CE uses a threshold function on the CDF.


## Specific Comments and Questions ##

1. Cakewalk is *very* closely related to the cross-entropy method. The authors acknowledge this connection, but I think they should begin by introducing CE and then explain how Cakewalk generalizes it. Both Cakewalk and CE approximate the reward CDF from K samples and use this to construct a surrogate reward. The difference is that Cakewalk uses the CDF directly, while CE uses a threshold function on the CDF.
2. The distribution proposed in section 3.2 assumes independence between the elements $x_j$. This seems problematic for some relatively simple problems. Consider $x$ a binary vector and reward equal to the parity $S(x) = \sum{x_j} % 2$.
3. In the experiments, there are large discrepancies between different optimizers on Cakewalk (e.g. SGA vs AdaGrad, Table 4). Is there any explanation for this?
4. How were the hyperparameters (learning rate, AdaGrad $\delta$, Adam $\beta_1, \beta_2$) chosen? It seems like a large assumption that the same learning rate would work for different methods, especially when some of them are normalizing the objective function. I would suggest tuning these values for each method independently.
5. It would be nice to see experimental results on more than one problem. The authors discuss their results on k-medoids in the appendices, but it seems like these results aren't quite complete yet.
6. In Table 3, the figure in bold is not the lowest (best) in the table. The reason for this is only given in a single sentence at the end of Section 6, so it is a little confusing. I would replace these values with N/A or something similar.

---

> ### Author Response · Authors · 2018-11-15
> **Response to issues raised by reviewer 2 - Part II**
>
> Except for the learning rate, all the hyper-parameters were chosen according to the values suggested by the authors of AdaGrad, and Adam. The learning rate was chosen as 1/K, with K=100 being the number of examples used to estimate the CDF. As our stated goal is to present an algorithm which can be blindly applied with some fixed set of hyper-parameters to any possible objective, one of the goals of the experiments is to show that in such a setting some methods will work, while others will fail. Thus, as a controlled experiment for this hypothesis, we first fixed the set of all hyper-parameters for all methods, and then proceeded to apply them to various problems. In this setting therefore, tuning the learning rate or any other hyper-parameter for that matter will compromise the validity of our results.
>
> Regarding table 3, we accept the reviewer’s suggestion, this is a good point. We particularly like the suggestion of writing NA or some such value, and we will use it to correct the paper.

---

> ### Author Response · Authors · 2018-11-15
> **Response to issues raised by reviewer 2 - Part I**
>
> We thank the reviewer for the evaluation. Please see our detailed response to several recurring issues at https://openreview.net/forum?id=Hkx-ii05FQ&noteId=HygFbNmL6X. In that response we address the following issues:
> (1) We emphasize fundamental differences between Cakewalk and CE. These go beyond the differences the reviewer mentions.
> (2) How the sampling distribution should not be considered as a part of Cakewalk, and that it is mostly provided as an example, and a basis for the reported experiments.
> (3) The experiments include results two tasks. Nonetheless, it appears the paper doesn’t convey this clearly, and we suggest two possible ways how to update the paper in this regard.
>
> Next, we try to answer the specific issues the reviewer mentions. First, we address the suggestion of introducing Cakewalk as a generalization of CE. While we were writing the paper we in fact considered presenting Cakewalk as the reviewer suggests. We eventually decided against this approach as CE is a method for adapting an importance sampler, and its convergence guarantees only apply when it is treated as such. The convergence guarantees of REINFORCE on the other hand still apply under our surrogate objective framework. This property allows us to explore various surrogates, where one such construction allows us to interpret CE as a policy gradient method, and another makes the basis for Cakewalk.
>
> Second, we address the issue of using a sampling distribution that assumes independence between the different dimensions. As the author correctly states, such a distribution will not always be useful, and one can design a problem for which this distribution will lead to a poor local optimum. Note however that a global maximizer for the objective suggested by the reviewer can be easily found just by random sampling: sampling such a maximizer has the same probably as sampling an odd integer - half. Nonetheless, for the clique problem such a distribution can be effective. Intuitively, if some node i is part of a large clique, then sampling x_i=1 is likely to result in a good objective as there are many nodes that are connected to i, and the chance of not sampling any of them decreases with the clique size. In this way, over time the probability for sampling such nodes becomes higher, and the chance of sampling all of them together increases. A similar reasoning applies for the k-medoids problem. We note that these kind of factorized distributions have a long history of being useful in machine learning. In a similar context to the one studied in the paper, such distributions have been studied by Rubinstein in his paper which discusses CE as an algorithm for combinatorial optimization, and in the classical bandit papers Exp3 is applied independently to several dimensions to study game theoretic problems. In different contexts, such distributions have also been used as naive mean field approximations in variational inference.
>
> Next, we address the question regarding the gradient update types. One intuitive explanation for why an algorithm that maintains a ‘memory’ of previous gradient updates like AdaGrad or Adam is required is that they protect against sampling biases. Consider for example the case when the execution is at the start, and the sampling distribution still has maximum entropy. Due to the combinatorial nature of the solution space, the examples that have been sampled thus far create a distorted representation of the solution space. In this case we could get that some x_i=j will occur few times, while some other x_k will not receive the value j at all. Now if we apply vanilla gradient updates this can skew the sampling distribution in random directions. Gradient updates such as those of AdaGrad and Adam on the other hand will lessen the impact of such deviations as the importance of each case is inversely proportional to the number of previous observations. As such deviations will inevitably occur whenever we rely on polynomially sized samples to represent a combinatorial solution space, without such corrections a gradient based adaptive sampling algorithm will almost surely fail. Indeed, as can be seen in tables 1,2 and 4, SGA almost never leads to a locally optimal solution. Furthermore, this reasoning explains why AdaGrad is superior to Adam: AdaGrad corrects against sampling biases that entail all the examples that have been encountered, while Adam does this only within some exponentially moving time window. Indeed, this phenomenon is studied in detail in the AdaGrad paper (though without assuming a data distribution), and sparse data like ours (one can say our data points are N indicator vectors of length M) is the first motivating example in their paper.

---

### Official Review · AnonReviewer1 · 2018-11-04
**Interesting approach but unconvincing experiments and motivation.**

**Rating:** 4
**Confidence:** 4

**Review:**

The authors argue that not knowing the distribution of rewards observed in the policy gradient algorithm hinders learning (and the tuning process). They propose to replace the reward term in the policy gradient algorithm with its centered empirical cumulative distribution, which has a fixed and known U[-1, 1] distribution. They test their methods on a toy task that consists in finding inclusion maximal cliques (which tests for local optimality) against REINFORCE (including their variants: centering the rewards with a mean baseline or normalizing them), the cross-entropy method and Exp3.

I think that the current draft lacks strong experimental results to properly demonstrate the usefulness of the method. The method is only evaluated on a single task and many confounding variables (the design of the reward function, factorizing the parametric distribution into marginals, reporting results for a single (non-tuned?) learning rate, etc.) make evaluation difficult. The usefulness of the approach is also lessened by the greater importance of the choice of the optimizer. I would like the method to be applied on other domains such as continuous non-convex optimization and reinforcement learning.

Additionally, I find the motivation for caring about local optimality unconvincing. I take exception that people care more about local optimality than the actual objective. From a practical point of view, local optimality is a mean (that can be achieved via heuristic algorithms) to an end (the objective itself). This also holds for k-means, which is usually run multiple times with different starting conditions.

Some comments:
- Table 3 is a bit confusing as-is (lower is only better when controlling on the quality of the best sample. e.g: REINFORCE has lower best-sample to total-sample ratio but its solutions are worse)
- It isn't clear from the tables that OCE_0.1 outperforms REINFORCE_Z (as is mentioned in the discussion).
- The paper should refer to 1) the reward shaping literature, 2) the growing line of works concerned with control variates for REINFORCE (such as VIMCO, MuProp, REBAR) and 3) the growing line of works concerned about combinatorial optimization with reinforcement learning (Neural Combinatorial Optimization with Reinforcement Learning, etc.)
- I would also encourage the authors to come up with a more descriptive name for the approach.

---

> ### Author Response · Authors · 2018-11-15
> **Response to issues raised by reviewer 1**
>
> We thank the reviewer for their evaluation. Please see our response at https://openreview.net/forum?id=Hkx-ii05FQ&noteId=HygFbNmL6X, where we also discuss our experimental framework. Even though we present results on two tasks, it appears the paper structure doesn’t convey this clearly, and we suggest two possible ways how to update the paper in this regard. We note in this context that even though we would also like to see Cakewalk evaluated on the domains mentioned by the reviewer, these are not part of our own research agenda, and accordingly our suggestions refer to other problems in combinatorial optimization. Next, we address other issues raised by the reviewer.
>
> First, we’d like to emphasize that the clique problem studied in the paper is far from being a toy problem. All the algorithms are evaluated on the DIMACS clique dataset which was published as part of the second DIMACS challenge which specifically focused on combinatorial optimization. Over the years, this dataset has become a standard benchmark for clique finding algorithms, and results on it are regularly published. In this respect, this dataset is an important benchmark for clique algorithms very much like CIFAR10 and CIFAR100 are for image classification methods. Notably, Cakewalk approaches the performance of the best clique finding algorithms that directly search a graph, and which are tailored to this specific task. Note that none of the tested methods were given enough samples even to recover the graph itself, as most graphs have more than 100 nodes, and we’ve allowed only for 100 |V| samples in each execution. To us this seems as a rather challenging setup, not just for the algorithms we’ve tested in this paper, but for any clique finding algorithm.
>
> Next, we wonder how would the reviewer correct the confounds mentioned with regard to the clique experiment. Providing a controlled experiment is always challenging, though the elements mentioned by the reviewer were specifically selected as to reduce various confounds. The main research question we try to address is whether algorithms that only rely on function evaluations can recover locally optimal solutions. Since the objective is the only source of information for such algorithms, an all-or-none kind of objective would not be very useful. Instead, the objective is designed in a manner that provides information even for partial solutions, thus allowing the tested algorithms to gradually improve the objective. In terms of the sampling distribution, as our focus is on the update step, we decided to use the simplest possible sampling distribution we can think of. In such a regime, we can attribute any performance gains to the algorithms themselves, and not to any prior knowledge that is reflected by the structure of some complex sampling distribution.
>
> Next, we agree that local optimality is a mean rather than a goal (the objective itself). Nonetheless, as in the problems we seek to address the global optimum cannot be found in polynomial time, the second best approach is first to design a method that can recover locally optimal solutions. Once such a method is available, repeated applications of that method can allow one to select a good solution, very much like the standard practice of repeated applications of k-means which the reviewer mentions. This reasoning however is dependent on a method’s capability of recovering locally optimal solutions, and therefore studying this ability makes for a worthwhile effort.
>
> Answers to the last comments:
> - Table 3 is indeed confusing, this is a good point. We will correct it.
> - Methods that apply a surrogate objective work best with AdaGrad. In this case, our data is a classical use which is explored in the AdaGrad paper uses as a motivating example. Not surprisingly, both Cakewalk and OCE work best with it. REINFORCE however is sensitive to the objective values, and it appears that Adam somewhat mitigates this problem. However, this is not as effective as applying a surrogate objective, and REINF_Z with Adam is outperformed by OCE_0.1 with AdaGrad in all measures.
> - Our frame of reference were algorithms that could be applied to any combinatorial problem, and which only rely on function evaluations. Control variates and reward shaping methods are mostly useful when tied to the particularities of a given objective, and thus do not fall into this category. In neural combinatorial optimization the study is focused on designing a sampling distribution that reflects some prior knowledge about a problem, and thus, we consider this line of work as orthogonal to ours. Having said that, we see how these areas might seem related, and we will revise the related work section to better emphasize the aforementioned differences.
> - We selected the name ‘Cakewalk’ after consulting with a few colleagues. Following a joint discussion, we concluded that this name has the best chance for increasing our work’s impact.

---

### Official Review · AnonReviewer3 · 2018-11-06
**Interesting paper with similar prior work and unconvincing experimental evaluation**

**Rating:** 5
**Confidence:** 3

**Review:**

The paper proposes an approach to construct surrogate objectives for the effective application of policy gradient methods to combinatorial optimization without known neighborhood structure. The surrogate is constructed with the goal of reducing the need of hyper-parameter tuning and evaluated on a clique finding task.

The proposed approach is very similar to the CE method by Rubinstein (as stated by the authors in the related work section), limiting the contributions of this paper. The proposed sampling distributions assumes independence between the random variables over which the authors optimize — I find it surprising that this leads to good empirical results are relatively little structure can be captured using this distribution. Can the authors elaborate on this? However, as also observed by the authors, the sampling distribution can also be replaced by more sophisticated distributions. The empirical evaluation is limited in considering only one task (clique finding), and the results seem to be quite sensitive to the chosen optimizer.

---

> ### Author Response · Authors · 2018-11-15
> **Response to issues raised by reviewer 3**
>
> We thank the reviewer for the evaluation. Please see our detailed response to several recurring issues at https://openreview.net/forum?id=Hkx-ii05FQ&noteId=HygFbNmL6X. In that response we address the following issues:
> (1) We emphasize fundamental differences between Cakewalk and CE.
> (2) How the sampling distribution should not be considered as a part of Cakewalk, and that it is mostly provided as an example, and a basis for the reported experiments.
> (3) The experiments include results two tasks. Nonetheless, it appears the paper doesn’t convey this clearly, and we suggest two possible ways how to update the paper in this regard.
>
> Next, we’ll try to provide some intuition as to why a sampling distribution that assumes independence between the different dimensions can be useful in some cases. The simple explanation is that in some problems the conditional expectation of the objective given that some x_i=j is much better than for other values x_i=k. In such cases, for each dimension the algorithm will tend to sample values which are useful to many possible solutions. In the clique problem for example, if some node i is part of a large clique, then sampling x_i=1 is likely to result in a good objective as there are many nodes that are connected to i, and the chance of not sampling any of them decreases with the clique size. In this way, over time the probability for sampling such nodes becomes higher, and the chance of sampling all of them together increases.
> Lastly, we note that these kind of factorized distributions have a long history of being useful  in machine learning. In a similar context to the one studied in the paper, such distributions have been studied by Rubinstein in his paper which discusses CE as an algorithm for combinatorial optimization, and in the classical bandit papers Exp3 is applied independently to several dimensions to study game theoretic problems. In different contexts, such distributions have also been used as naive mean field approximations in variational inference.

---

### Author Response · Authors · 2018-11-11
**Responses to recurring concerns, and general comments**

We would like to thank the reviewers for their time and effort. We first address issues which have been raised by more than one reviewer, and we defer answers to specific questions to the correspondences with the reviewers.

It seems that the reviewers are concerned that the paper presents results only on a single task (reviewer 2 does mention the additional results supplied in the appendix, but these are only mentioned in passing). There is little doubt that reporting results on few problems provides stronger evidence in support of a method. For that reason, in addition to the results on clique-finding, we also report results on k-medoids clustering in the appendix. Each task is designed to make a different kind of comparison, and while the k-medoids problem allows us to compare Cakewalk to problem specific algorithms, it is the results on clique-finding which address the main research question we propose in the paper. In the last few years policy gradient methods have been applied to a considerable variety of problems. As some of these are essentially black-box combinatorial optimization problems, this trend made us wonder whether there was empirical evidence that verifies the optimality of their returned solutions. From what we could gather, it seemed to us that this question was not answered in a convincing manner. Thus, in the main body of the paper our goal was to provide a controlled experiment which addresses this question as it seemed to us that answers to this question will carry the most weight. We see however that this approach was not convincing to the reviewers, and thus, we propose to adjust the paper in one of the following two ways. We can restructure the paper to include the additional results on the k-medoids problem in the main body of the paper. Alternatively, we can provide another set of results on the TSP problem while using a similar setup to the one used for the clique problem. For TSP however, definitions of solutions’ neighborhoods are rather artificial, and therefore a study of local optimality will not make much sense. Such edits however will require us to extend the main body of the paper by 1 or 2 pages (the reason we decided not to include such results in the first place), but if the reviewers think such an update can significantly increase the paper’s ratings we will proceed to do so.

Next we address the similarity to the CE method. As noted in sections 3.1 and 4, our method is indeed closely related to CE. Nonetheless, there are fundamental differences that separate the two, and we would like to emphasize these in this context.
1) The CE is a batch method, and not an online algorithm such as Cakewalk or REINFORCE. In each iteration of CE a large sample is required, and thus CE’s iterations are considerably more computationally expensive than those of Cakewalk. In the paper we use our surrogate objective construction to interpret CE as a policy gradient method, and that in turn allows us to derive a new online version for CE which significantly deviates from Rubinstein’s algorithm. Furthermore, as discussed in section 3.1, this online version has major shortcomings, and these lead us to propose Cakewalk.
2) At the end of each iteration of CE, the sampling distribution’s parameters are updated to the maximum likelihood estimate (MLE) of the examples which achieved the best objective values at that iteration. Thus, in order to alleviate the computational costs of finding the MLE, the traditional approach is to use a sampling distribution for which this update can be done in closed form. In Cakewalk however, the parameters are gradually updated through gradient updates, a property which has has two advantages. First, in Cakewalk one can use any differentiable sampling distribution which has a bounded gradient, and thus, Cakewalk can be applied with considerably more versatile sampling distributions than CE. Second, by relying on bounded gradient updates, in Cakewalk the empirical CDF can be estimated in an online manner, leading to the computational savings mentioned in (1).
3) Lastly, in spite of any apparent similarities, the fact remains that Cakewalk and online CE behave differently on real data. Notably we show that these differences are statistically significant, and therefore these results provide objective evidence for the difference between the two that go beyond any subjective interpretation of similarity.

Having said that, we acknowledge that points (1) and (2) are missing from the related work section, and therefore we will update the paper to include them.

Lastly, we would like to emphasize that the sampling distribution is *not* part of the Cakewalk method. Cakewalk can be applied with various sampling distributions, and the specified distribution in section 3.2 was provided as part of the experimental setup. We see however how the structure of the paper is confusing in this respect, and we will update the paper to better emphasize this point.

---

### Author Response · Authors · 2018-11-25
**Revision**

Following our response to the major issues raised by the reviewers in comment https://openreview.net/forum?id=Hkx-ii05FQ&noteId=HygFbNmL6X , we have uploaded a new version of our paper.  We hope this version addresses the reviewers' major concerns. Specifically:
- We have edited the end of the introduction to better emphasize that we report results on two tasks: finding inclusion maximal cliques, and k-medoids clustering.
- In the related work section we have edited the part which discusses the relation to CE. We believe that the current version better emphasizes the fundamental differences between the two.
- We have edited the section that discusses the sampling distribution. In this version we try to better emphasize that Cakewalk can be applied with a wide variety of sampling distributions, and it is not tied to the particular example specified in the paper.

In addition:
- Table 3 was edited according to the useful suggestions of the reviewers.
- The algorithm's figure was corrected, the previous version had a small typo.
- Another recurring typo was corrected at the end of section B.
- The description of the analysis of the results on the k-medoids problem was edited. We hope the new version provides a clearer explanation of how we produced the algorithms' performance rankings.

---

### Meta-Review · Area_Chair1 · 2018-12-17
**Variation of "cross-entropy method", not considered sufficient for acceptance**

**Confidence:** 5
**Recommendation:** Reject

**Metareview:**

The paper investigates a variant of the "cross-entropy method" (CME) for heuristic combinatorial optimization, based on stochastically improving a search distribution via policy optimization in a surrogate objective.

Unfortunately, the reviewers unanimously recommended rejection, noting that the significance of the contribution over CME remains far from clear and insufficiently supported by the given evidence.  The experimental evaluation was unconvincing to all of the reviewers, particularly since only one artificial problem (clique finding) was considered in the paper (with an additional problem, k-medoid clustering, briefly and incompletely considered in the appendix).  Several additional concerns were raised about the experimental evaluation, which triggered lengthy author responses but really need to be properly handled in the paper itself:

- The sensitivity of performance to the optimization algorithm is a concern and requires more detailed understanding so that reasonable choices can be made in practice.

- The independence assumption between search components is an extreme simplification that limits the appeal and applicability of the proposed approach.  Even after author response, it remains unconvincing that an independent search distribution over subcomponents can be effective in challenging combinatorial spaces.  Concrete evidence on challenging problems would be a more effective evidence than discussion.

- The comparisons omitted any tailored algorithms for the specific problems.  Even if the authors insist on only comparing to more "general purpose" methods, there is a large space of evolutionary and Bayesian optimization strategies that have been neglected from the comparison.  A justification is needed for such an omission (if indeed it is even justifiable).